# Warm-starting Push-Relabel

**Sami Davies**[∗]**, Sergei Vassilvitskii**[†]**, Yuyan Wang**[†]
[∗]UC Berkeley, [†]Google Research – New York
davies@berkeley.edu, {sergeiv,wangyy}@google.com

## Abstract

Push-Relabel is one of the most celebrated network flow algorithms. Maintaining a pre-flow that saturates a cut, it enjoys better theoretical and empirical running time than other flow algorithms, such as Ford-Fulkerson. In practice, Push-Relabel is even faster than what theoretical guarantees can promise, in part because of the use of good heuristics for seeding and updating the iterative algorithm. However, it remains unclear how to run Push-Relabel on an arbitrary initialization that is not necessarily a pre-flow or cut-saturating. We provide the first theoretical guarantees for warm-starting Push-Relabel with a predicted flow, where our learning-augmented version benefits from fast running time when the predicted flow is close to an optimal flow, while maintaining robust worst-case guarantees. Interestingly, our algorithm uses the *gap relabeling heuristic*, which has long been employed in practice, even though prior to our work there was no rigorous theoretical justification for why it can lead to run-time improvements. We then provide experiments that show our warm-started Push-Relabel also works well in practice.

## 1 Introduction

Maximum flow is a fundamental problem in combinatorial optimization. It admits many algorithms, from the famous Ford-Fulkerson algorithm [FF56] which employs augmenting paths, to recent near-linear time scaling based approaches [CKL+22]. In practice, however, the *Push-Relabel* family of algorithms remains one of the most versatile and is often the standard benchmark to which new flow algorithms are compared [Wil19, CH09].

Designed by Goldberg and Tarjan [GT88], the core Push-Relabel algorithm (Algorithm 1) has running time $O(n^2m)$, where $n$ and $m$ are the number of vertices and edges in the network. Given the popularity of max-flow as a subroutine in large-scale applications [BK04, KBR07, Wil19], it is no surprise that improving running times has been a subject of a lot of study, with multiple heuristic methods being developed [JM+93, CG95, AKMO97, Gol08].

To complement these heuristics, researchers studied max-flow in the *algorithms with predictions* framework (or learning-augmented algorithms) [MV22]. Two groups initiated the study and proved that the running time of the Edmonds-Karp selection rule for Ford Fulkerson can be improved from $O(m^2n)$ to $O(m||f^* - \hat{f}||_1)$, where $f^*$ is an optimal flow on the network and $\hat{f}$ is a predicted flow [DMVW23, PZ22]. These algorithms start the augmenting path algorithms from a feasible flow obtained from the predicted flow, then bound the number of augmentations by the $\ell_1$-distance between the predicted and maximum flows.

While these works have improved upon the cold-start, non learning-augmented versions, it is important to note that they have improved upon sub-optimal algorithms for max flow. In this work, we show how to warm-start Push-Relabel, whose cold-start version is nearly state-of-the-art for the maximum flow problem. This directly addresses the challenge specified in [DMVW23] on bringing a rigorous analysis for warm-starting non-augmenting path style algorithms. In the process of doing so, we provide a theoretical explanation for the success of the popular *gap relabeling heuristic* in improving

38th Conference on Neural Information Processing Systems (NeurIPS 2024).

the running time of Push-Relabel algorithms. Specifically, both the gap relabeling heuristic and our algorithm maintain a cut with *monotonically decreasing t-side nodes* (see Section 1.2 for more), which directly leads to improved running times for our version of Push-Relabel and the gap relabeling heuristic. Lastly, we show that our theory is predictive of what happens in practice with experiments on the image segmentation problem.

## 1.1 Preliminaries

**Graph, flow and cut concepts.** Our input is a network $G = (V, E)$, where each directed edge $e \in E$ is equipped with an integral capacity $c_e \in \mathbb{Z}_{\geq 0}$. Let $|V| = n$ and $|E| = m$. $G$ contains nodes $s$, the source, and $t$, the sink. $G$ is connected: $\forall u \in V$, there are both $s - u$ and $u - t$ paths in $G$. A flow $f \in \mathbb{Z}_{\geq 0}^m$ is feasible if it satisfies: (1) *flow conservation*, meaning any $u \in V \setminus \{s, t\}$ satisfy $\sum_{(v,u) \in E} f_e = \sum_{(u,w) \in E} f_e$; (2) *capacity constraints*, meaning for all $e \in E$, $f_e \leq c_e$. Our goal is to find the maximum flow, i.e. one with the largest amount of flow leaving $s$.

We call $f$ a *pseudo-flow* if it satisfies capacity constraints only. A node $u \in V \setminus \{s, t\}$ is said to have *excess* if it has more incoming flow than outgoing, i.e., $\sum_{(v,u) \in E} f_e > \sum_{(u,w) \in E} f_e$; analogously it has *deficit* if its outgoing flow is more than ingoing. We denote the excess and deficit of a node $u$ with respect to $f$ as $\mathsf{exc}_f(u) = \max\{\sum_{(v,u) \in E} f_e - \sum_{(u,w) \in E} f_e, 0\}$ and $\mathsf{def}_f(u) = \max\{\sum_{(u,w) \in E} f_e - \sum_{(v,u) \in E} f_e, 0\}$, where at most one can be positive. A pseudo-flow can have both excesses and deficits, whereas a *pre-flow* is a pseudo-flow with excess only.

For a pseudo-flow $f$, the *residual graph* $G_f$ is a network on $V$; for every $e = (u, v) \in E$, $G_f$ has edge $e$ with capacity $c'_e = c_e - f_e$ and a backwards edge $(v, u)$ with capacity $f_e$. Let $E(G_f)$ denote the edges in $G_f$. The value of a pseudo-flow $f$ is $\mathsf{val}(f) = \sum_{e=(s,u)} f_e$, the total flow going out of $s$. Notice that this is not necessarily equivalent to the total flow into $t$ since flow conservation is not satisfied. A *cut-saturating* pseudo-flow is one that saturates some $s - t$ cut in the network. Push-Relabel maintains a cut-saturating pre-flow; equivalently, there is no $s - t$ path in the residual graph of the pre-flow. We use $\delta(S, T)$ to denote an $s - t$ cut between two sets $S$ and $T$. Note that the cut induced by any cut-saturating pseudo-flow $f$ can be found by taking $T = \{u \in V : \exists u - t \text{ path in } G_f\}$ (including $t$) and $S = V \setminus T$.

**Prediction.** The prediction that we will use to seed Push-Relabel is some $\hat{f} \in \mathbb{Z}_{\geq 0}^m$, which is a set of integral values on each edge. Observe that one can always cap the prediction by the capacity on every edge to maintain capacity constraints, so throughout this paper we will assume $\hat{f}$ is a pseudo-flow. It is important to note that our predicted flow is *not* necessarily feasible or cut-saturating, and part of the technical challenge is making use of a good predicted flow despite its infeasibility.

**Error metric.** We measure the error of a predicted pseudo-flow $\hat{f}$ on $G$. The smaller the error is, the higher quality the prediction is, and the less time Push-Relabel seeded with $\hat{f}$ should take. A pseudo-flow becomes a maximum flow when it is both feasible and cut-saturating. Hence, the error measures how far $\hat{f}$ is from being cut-saturating while being feasible. We say that a pseudo-flow $\hat{f}$ is $\sigma$ *far from being cut-saturating* if there exists a feasible flow $f'$ on $G_{\hat{f}}$ where $\mathsf{val}(f') \leq \sigma$ and $\hat{f} + f'$ is cut-saturating on $G$ (though the cut does not have to be a min-cut). To measure how far $\hat{f}$ is from being feasible, we sum up the total excesses and deficits. In total we use the following error metric:

**Definition 1.** *For pseudo-flow $\hat{f}$ on network $G$, the* error *of $\hat{f}$ is the smallest integer $\eta$ such that (1) $\hat{f}$ is $\eta$ far from being cut-saturating and (2) $\sum_{u \in V \setminus \{s,t\}} \mathsf{exc}_{\hat{f}}(u) + \mathsf{def}_{\hat{f}}(u) \leq \eta$.*

If $\eta = 0$, $\hat{f}$ is the max-flow and the cut that is saturated is the min-cut. The previously studied error metric for predicted flows, such as by [DMVW23] and [PZ22], was $||f^* - \hat{f}||_1$, for any max-flow $f^*$.

PAC-learnability is the standard to justify that the choice of prediction and error metric are reasonable. Flows are PAC-learnable with respect to the $\ell_1$-norm [DMVW23]. Our results hold replacing our error metric with the $\ell_1$-norm because our metric provides a more fine-grained guarantee than the $\ell_1$-norm (i.e., if a prediction $\hat{f}$ has error $\eta$, then $\eta \leq ||f^* - \hat{f}||_1$). Thus we can omit any theoretical discussion of learnability. We present this work with respect to our error metric as we find the $\ell_1$ error metric to be unintuitive, as it is not really very descriptive of how good a predicted flow is.

**Push-Relabel.** The "vanilla" Push-Relabel algorithm maintains a pre-flow and set of valid *heights* (also called labels) on nodes. Heights $h : V \to \mathbb{Z}_{\geq 0}$ are *valid* for $f$ if for every edge $(u, v) \in E(G_f)$ with positive capacity, $h(u) \leq h(v) + 1$, and if $h(s) = n$ and $h(t) = 0$. An edge $(u, v) \in E(G_f)$ is called *admissible* if $h(u) = h(v) + 1$ and $c'_{(u,v)} > 0$, which means we can push flow from $u$ to $v$. The algorithm starts with $f^{\text{init}}$, where $f_e^{\text{init}} = c_e$ for all $e = (s, u)$ and otherwise $f_e^{\text{init}} = 0$. It then pushes flow on admissible edges $(u, v)$ for $u$ with excess flow when possible, and otherwise the algorithm updates the heights. See Algorithm 1.

---

**Algorithm 1** Push-Relabel

> **Input**: Network $G$
> Define $f_e = c_e$ for $e = (s, u)$ and $f_e = 0$ for all other $e$
> Define $h(u) = 0$ for all $u \in V \setminus \{s\}$ and $h(s) = n$
> Build residual network $G_f$
> **while** $\exists$ node $u$ with $\mathsf{exc}_f(u) > 0$ **do**
>      **if** $\exists$ admissible $(u, v) \in E(G_f)$ with $f_{(u,v)} < c_{(u,v)}$ **then**
>          Update $f$ by sending an additional flow value of $\min\{\mathsf{exc}_f(u), c'_{(u,v)}\}$ along $(u, v)$
>          Update $G_f$ accordingly
>      **else** update $h(u) = 1 + \min_{v:(u,v) \in E(G_f)} h(v)$
> **Output**: $f$

---

All heights in Push-Relabel are bounded.

**Lemma 1.** *For a pre-flow $f$ on network $G$, every node $u$ with $\mathsf{exc}_f(u) > 0$ has a path in $G_f$ to $s$. Further, for $d(u, v)$ the length of the shortest path from $u$ to $v$ in $G_f$, any valid heights in Push-Relabel (Algorithm 1) satisfy $h(u) \leq h(v) + d(u, v)$. For all $u \in V$, $h(u) \leq h(s) + n = 2n$.*

At any point of the algorithm, the $s - t$ cut maintained can be found using the heights.

**Lemma 2.** *For a pseudo-flow $f$ with valid heights $h$ on network $G$, let $\theta$ be the smallest positive integer such that $\theta \notin \{h(u)\}_{u \in V}$. Then sets $S = \{u \in V : h(u) > \theta\}$ and $T = \{u \in V : h(u) < \theta\}$ form a cut saturated by $f$.*

We call this cut *induced by the heights*. Indeed, such a threshold $\theta$ can be found because $\{h(u)\}_{u \in V}$ has at most $n$ different values, but $h(s) = n$ and $h(t) = 0$, so among the $n + 1$ values $\{0, 1, \ldots, n\}$, at least one is not in the set. Note that $\delta(S, T)$ is a saturated cut by definition of valid heights. A saturated cut can also be defined by whether a node can reach the sink in the residual graph.

**Lemma 3.** *For any pseudo-flow $f$ on network $G$, let $T$ be all nodes that can reach $t$ in $G_f$ (including $t$) and $S = V \setminus T$. If $s \in S$, then $\delta(S, T)$ is a saturated $s - t$ cut.*

Lemmas 2 and 3 apply to all pseudo-flows, whereas vanilla Push-Relabel must take a pre-flow as input. Before this work, it was unclear how to seed Push-Relabel with anything other than $f^{\text{init}}$.

With the *gap relabeling heuristic*, whenever there is some integer $0 < \theta < n$ with no nodes at height $\theta$, then nodes with height between $\theta$ and $n$ have their height increased to $n$. See Algorithm 2.

## 1.2 Technical contribution

We first study Push-Relabel with the gap relabeling heuristic when seeded with a prediction that is a cut-saturating pre-flow with error $\eta$. We prove Algorithm 2 finds an optimal solution in time $O(\eta \cdot n^2)$. The running time also holds for cold-start versions of the algorithm when the max-flow/min-cut value is known to be bounded by $\eta$. This is (1) the first theoretical analysis of the gap relabeling heuristic, and, (2) the first result showing a running time bounded by the volume of the cut in Push-Relabel. While Ford-Fulkerson admits a naive running time bound of $O(\eta \cdot m)$ when the max-flow value is at most $\eta$, an analogous claim was not previously known for Push-Relabel.

The algorithm maintains a cut whose $t$-side is *monotonically decreasing* (i.e., it moves nodes on the $t$-side of the cut to the $s$-side, but not the other way around), and resolves excess on the $t$-side by routing excess flow to $t$, or updating the cut so the excess node is on the $s$-side of the new cut. The same insight will be used in our warm-started version of Push-Relabel seeded with any pseudo-flow.

Our main result applies in the general setting where the prediction is any pseudo-flow.

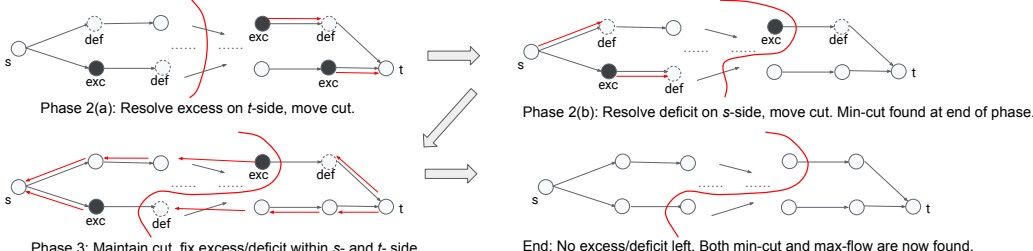

Figure 1: An illustration of warm-start Push-Relabel, seeded with a cut-saturating pseudo-flow. The red curve denotes the cut. The black arrows denote the existing flows, whereas the red arrows denote the flows sent in each phase to resolve excess/deficits.

**Theorem 1.** *Given a predicted pseudo-flow $\hat{f}$ with error $\eta$ on network $G$, there exists a warm-start version of Push-Relabel that obtains the minimum cut in time $O(\eta \cdot n^2)$.*

Our warm-start version of Push-Relabel that is seeded with a general pseudo-flow has several phases. In the first phase, we show that one can modify the prediction to be a cut-saturating pseudo-flow. We begin the second phase by routing flow *within* the two sides of the cut induced by the pseudo-flow to resolve some of the excess and deficit. The maintained cut gradually changes as we send flow from node to node, and push certain nodes to different sides of the cut. We continue changing the cut until all excess nodes end up on the $s$-side of the cut and all deficit nodes end up on the $t$-side of the cut. This "swapping" procedure between excess and deficits nodes between the $s$- and $t$- sides of the cut is our biggest technical innovation. Either the excesses are resolved within the $t$-side of the cut, or we find a new cut between the $t$-side excess nodes and the $t$-side deficit nodes plus $t$. We modify the cut in $G$ accordingly to separate all excess from the $t$-side, which also results in a cut whose $t$-side is monotonically decreasing—an interesting point which also occurs in the cut maintained by the gap relabeling heuristic. A mirrored version of this process is performed on the $s$-side of the cut.

In the final phase, we have a new cut-saturating pseudo-flow with all excess nodes on the $s$-side of the cut and all deficit nodes on the $t$-side of the cut. This cut is actually a min-cut. On the $s$-side, the excess nodes send flow to the source, and on the $t$-side, the sink sends flow to deficit nodes (hence removing existing flows). The result is a max-flow. See Figure 1 for an illustration of phases 2 and 3.

In Section 4, we run our warm-start Push-Relabel compared to a cold-start version. We see that the warm-start improves over the cold-start by a larger percentage as the size of the image increases.

Our work identifies that there is a monotonic property in the cuts when Push-Relabel is implemented with the gap relabeling heuristic that directly results in faster running time when the value of the max flow/min cut is small (Corollary 1). While such a result was known for Ford-Fulkerson, this was *not* known for Push-Relabel prior to our work. We leave as an open direction whether this monotonic property may have further theoretical importance in improving Push-Relabel analyses on more general classes of networks.

The running time of warm-start Push-Relabel is $O(n^2\eta)$, while the running time of warm-start Edmonds-Karp is $O(m\eta)$ [DMVW23]. We do not believe that the bound for warm-start Push-Relabel being no faster than that of warm-start Edmonds-Karp is a weakness in our analysis, but is instead an artifact of the community's lack of understanding on why Push-Relabel performs so much better in practice than its theoretical bounds guarantee. Improving our theoretical understanding of algorithms that do well in practice (like Push-Relabel or the Simplex Method) is one of the pillars of the field of beyond worst-case analysis [Rou21]. Researchers have long suspected that there should be a way to parameterize instances or use practical heuristics to justify why Push-Relabel is so much better in practice than its theoretical worst-case bound would suggest [CG95, Hoc08, CM99].

## 2 Gap Relabeling Push-Relabel: Cold- and Warm-Start

We briefly overview the performance of Push-Relabel with the gap relabeling heuristic (Algorithm 2) when given a cut-saturating *pre-flow* $f$, and tie the running time to the error of $f$. Proofs in this section are deferred to Appendix A.2, as their more involved analogs are in Section 3.

---

**Algorithm 2** Warm-start Push-Relabel with Gap Relabeling

---

**Input:** Network $G$, a cut-saturating pre-flow $f$
Construct residual network $G_f$ with capacity $c'$
Run Algorithm 4 (see Appendix A.1) on $G$ and $f$, obtain valid heights $h$ and initial cut $(S, T)$
Initialize $\theta = \min\{z \in \mathbb{Z}_{>0} : \nexists u \in T \text{ with } z = h(u)\}$
**while** $\exists u \in T$ with $\text{exc}_f(u) > 0$ **do**
 **if** $\exists v \in T$ such that $(u, v)$ is admissible in $G_f$ **then**
  Update $f$ by sending an additional flow value of $\min\{\text{exc}_f(u), c'_{(u,v)}\}$ along $(u, v)$
  Update $G_f$ accordingly
 **else**
  Raise the height of $u$ to be $h(u) = \min_{v:(u,v)} h(v) + 1$
  Update $\theta = \min\{z \in \mathbb{Z}_{>0} : \nexists u \in T \text{ with } z = h(u)\}$
  **for** $p \in T$ with $h(p) > \theta$ **do**
   Remove $p$ from $T$, add $p$ to $S$
   Update $p$'s height to $h(p) = n$
Take $G_f$ as input and run Algorithm 1 on it to fix excesses, outputs flow $f^*$
Return flow $f + f^*$ and the cut $\delta(S, T)$ it maintains

---

Algorithm 2 begins by running Algorithm 4 as a subroutine to find the cut saturated by $f$ and define valid heights for $f$ which also induce that cut. Algorithm 4 runs a BFS in the residual graph to find all nodes that have a path to $t$ and names this set $T$. The other nodes belong to $S$. By Lemma 3 the cut $\delta(S, T)$ is saturated by $f$. In the main WHILE loop, the algorithm maintains the cut $\delta(S, T)$ such that all heights in $T$ compose a series of *consecutive* numbers starting from 0. The cut only changes when a node is relabeled in a way that results in a break in the series of consecutive heights starting from 0 in $T$, where the smallest missing height (the node's height before relabeling) is denoted by $\theta$. The algorithm then removes all nodes from $T$ with height bigger than $\theta$; importantly, these nodes will *never* enter $T$ again. The WHILE loop terminates when $T$ has no excess, thus finding the min cut.

**Lemma 4.** *Let $f$ be a pre-flow saturating cut $\delta(S, T)$ on network $G$. If there are no excess nodes in $T$, then all excess in $S$ can be sent back to $s$ without crossing the cut, implying $\delta(S, T)$ is a min-cut.*

We show that the running time is tied to $\eta$, which, in this case, is the total excess in $f$.

**Theorem 2.** *Given a cut-saturating pre-flow $f$ with error $\eta$ on network $G$, Algorithm 2 finds a max-flow/min-cut in running time $O(\eta \cdot n^2)$.*

Although Algorithm 2 is presented as being seeded with an existing pre-flow, the same bound applies to the cold-start gap relabeling Push-Relabel when the min-cut of $G$ is at most $\eta$. One only has to seed it with $f^{\text{init}}$ as in Algorithm 1, which saturates the cut $\delta(\{s\}, V \setminus \{s\})$. This will be useful in Section 3, as we repeatedly use Algorithm 2 as a subroutine on networks with small min-cut.

**Corollary 1.** *If network $G$ is known to have a max-flow/min-cut value of at most $\eta$, one can use Algorithm 2 to obtain a max-flow and min-cut for $G$ in running time $O(\eta \cdot n^2)$.*

## 3 Warm-starting Push-Relabel with General Pseudo-flows

We extend the results in Section 2 to when the given prediction is a general pseudo-flow $\hat{f}$ as opposed to a cut-saturating pre-flow, i.e., $\hat{f}$ may not be cut-saturating and may have deficit nodes. The first phase of our algorithm computes $\eta$ and augments $\hat{f}$ by finding an $s - t$ flow to add to $\hat{f}$ so that the resulting pseudo-flow saturates a cut. We defer discussion of this pre-processing phase to Appendix A.2 (see Lemma 8 and Algorithm 5). Once we have a cut-saturating pseudo-flow $f$ and $\eta$, we are ready to define the accompanying heights and cut using Algorithm 4. Note that the initial cut with two sides $T_0 = \{u \in V : \exists u - t \text{ path in } G_f\}$ and $S_0 = V \setminus T_0$ is by definition the same cut as that induced by the heights (as in Lemma 2).

We update the pseudo-flow so that it always maintains a saturated cut, but eventually, the nodes with excess and the nodes with deficit are separated by the saturated cut. This is a generalization of what happens in Algorithm 2, where we transfer all excess nodes to the $s$-side of the cut. Here, we transfer

---

**Algorithm 3** Moving all excess to the $s$-side of the cut

---

**Input**: Network $G$, a cut saturating pseudo-flow $f$, and error $\eta$
Run Algorithm 4, get output heights $h$
Let $T_0 = \{u \in V : \exists u - t \text{ path in } G_f\}$ and $S_0 = V \setminus T_0$
Build the residual $G_f$
Build $G'$ on copy of $G_f[T_0]$ plus $\{s^*, t^*\}$
**for** excess node $u \in T_0 \setminus \{t\}$ **do**
    Add edge $(s^*, u)$ with capacity $\mathsf{exc}_f(u)$
**for** deficit node $v \in T_0$ **do**
    Add edge $(v, t^*)$ with capacity $\mathsf{def}_f(u)$
Add edge $(t, t^*)$ with capacity $\eta + 1$
Let $f^{\mathsf{init}}_{(s^*, u)} = c_{(s^*, u)}$ for all $(s^*, u)$, and all other $f^{\mathsf{init}}_e = 0$
Run Algorithm 2 on $G'$ and $f^{\mathsf{init}}$, outputs $f'$ and $T_0', T_0''$
**for** all copies of $e = (u, v) \in E(G_f)$ where $f'_e > 0$ **do**
    Update $f_e \leftarrow f_e + f'_e$
**Output**: Flow $f$ and cut parts $S_0 \cup T_0'$ and $T_0''$

---

all excess to the $s$-side, and all deficit to the $t$-side of the cut. Interestingly, we observe that this is the sufficient condition for the pseudo-flow to saturate a min-cut. Lemma 5 extends Lemma 4.

**Lemma 5.** *For a cut-saturating pseudo-flow $f$ for a network $G$, let $\delta(S, T)$ be a cut it saturates. If all nodes in $T$ have no excess and all nodes in $S$ have no deficit, then the cut is a minimum cut.*

To prove Lemma 5, we use the following result from [DMVW23]:

**Lemma 6** (Lemma 5 [DMVW23])**.** *Given any pseudo-flow $f$ for $G$, every excess node has a path in $G_f$ to either a deficit node or $s$; every deficit node has a path in $G_f$ from either an excess node or $t$.*

*Proof of Lemma 5.* In $G_f$, by Lemma 6, every excess node $u$ in $S$ must have a path to either a deficit node or $s$. Since $f$ currently saturates $\delta(S, T)$, the path cannot go across this cut and reach $T$, where all the deficits are. Therefore, $u$ has a path back to $s$ *within* set $S$. Similarly, for every deficit node $v \in T$ there is a path that starts with either an excess node or $t$ and ends with $v$. Again, since the cut $\delta(S, T)$ is already saturated, the path must be within $T$, hence can only be from $t$ to $v$. It follows that we can send all excess to $s$ and send flow from $t$ to all deficit nodes until the pseudo-flow becomes a feasible flow. Notice that $\delta(S, T)$ remains saturated in this process, hence the resulting flow is max-flow and $\delta(S, T)$ is min-cut. $\qquad\square$

By Lemma 5, it is sufficient to find a pseudo-flow and accompanying saturated cut where the excess nodes are all on the $s$-side and the deficit nodes are all on the $t$-side. We focus on the $t$-side of the cut, and show that the same can be done for the $s$-side by considering the backwards network.

**Moving excess to the $s$-side.** To resolve excess on the $t$-side, we solve an auxiliary graph problem. The goal is to send the maximum amount of flow from excess nodes to either deficit nodes or $t$ *within* the $t$-side (currently denoted $T_0$). If the max-flow in this problem matches the total excess in $T_0$, all excess can be resolved locally and only deficits remain; otherwise, the max-flow solution on the auxiliary graph also provides us with a min-cut that "blocks" excess nodes from deficit nodes and $t$. This cut will become the new cut maintained by the pseudo-flow after adding the auxiliary flow to it.

To construct the auxiliary $G'$, take the induced subgraph $G_f[T_0]$, and add a super-source and -sink $s^*$ and $t^*$ to it. Add edges $(s^*, u)$ with capacity $\mathsf{exc}_f(u)$ for every excess node $u \in T_0$; add edges $(v, t^*)$ with capacity $\mathsf{def}_f(v)$ for every deficit node $v \in T_0$; and add an edge $(t, t^*)$ with capacity $\eta + 1$.

When we run cold-start Push-Relabel (Algorithm 2) on $G'$, it outputs a flow $f'$ and the $s^* - t^*$ cut $\delta(T_0', T_0'')$. Note that $t \in T_0''$, since $(t, t^*)$ has capacity $\eta + 1$ and cannot be in the min cut which is bounded by $\eta$ by definition. Any $s^* - t^*$ path $p$ in $G'$ along which $f$ sends $\delta$ units of flow exactly identifies nodes $u$ and $v$ (where $(s^*, u) \in p$ and $(v, t^*) \in p$) for which $\delta$ units of flow can be sent from $u$ to $v$ along the interior of $p$ in $G_f$. We can send flow as indicated by $f'$ to update $f$ (Algorithm 3). We obtain the following guarantee.

**Claim 1.** *In Algorithm 3, the output pseudo-flow $f$ saturates the cut $\delta(S_0 \cup T_0', T_0'')$, and all excess nodes are in $S_0 \cup T_0'$. Moreover, the total excess and deficit in $G$ has not increased.*

*Proof of Claim 1.* Let $f_{\mathsf{old}}$ denote the input to Algorithm 3. The fact that the output $f$ saturates the cut $\delta(S_0 \cup T_0', T_0'')$ immediately follows from the fact that $f'$ saturated the cut $\delta(T_0', T_0'')$ in $G'$. Indeed, all edges from $T_0'$ to $T_0''$ are now saturated and all edges from $T_0''$ to $T_0'$ have no flow. All edges from $S_0$ to $T_0''$ are already saturated in the old flow $f_{\mathsf{old}}$ and remain so after adding $f'$ since its flows are locally within $T_0$. For the same reason, all edges from $T_0''$ back to $S_0$ still have no flow.

Now, we consider the total excess and deficit. First note that the nodes that have excess/deficit with respect to the updated pseudo-flow $f$ are a subset of the nodes that had excess/deficit with respect to $f_{\mathsf{old}}$, and the excess/deficit of a node is clearly never increased. Assume for sake of contradiction there is an excess node $u \in T_0''$. Then $u$ had excess with respect to $f_{\mathsf{old}}$ too, so there is an edge $(s^*, u)$ that had capacity $\mathsf{exc}_{f_{\mathsf{old}}}(u)$ in $G'$ but was not saturated by $f'$. Further, since a min-cut in $G'$ is $\delta(T_0', T_0'')$, it must be that $u$ can reach $t$ in $G'$. This means that in $G'$ there is a path with positive remaining capacity between $s^*$ and $t^*$, contradicting the fact that $f'$ was a max-flow in $G'$. □

By Claim 1, the updated $f$ satisfies the conditions of Lemma 7 by taking $S^* = S_0 \cup T_0'$ and $T^* = T_0''$. The running time in Lemma 7 follows by applying Corollary 1 on $G'$. Thus we have the following.

**Lemma 7.** *Let $f$ be a pseudo-flow for network $G$ with error $\eta$ saturating cut $\delta(S_0, T_0)$. Algorithm 3 finds a new cut-saturating pseudo-flow in time $O(\eta \cdot n^2)$ that (i) saturates an additional $s - t$ cut $\delta(S^*, T^*)$, (ii) has no excess nodes in $T^*$, and (iii) has total excess and deficit still bounded by $\eta$.*

We can do a similar procedure for the $s$-side of the cut, this time removing deficit nodes. This is the backward process of what happens to the $t$-side, and can be done by reversing the graph edges and flows and running Algorithm 3 on the reversed network; see Appendix A.2 and Algorithm 6.

**Corollary 2.** *Let $f$ be a pseudo-flow for network $G$ with error $\eta$ that saturates cut $\delta(S_0, T_0)$. One can update $f$ in time $O(\eta \cdot n^2)$ so that all flow in $T_0$ remains unchanged, but now $f$ saturates a cut $\delta(S^*, T^*)$ and there are no deficit nodes in $S^*$.*

**Coping with unknown $\eta$** Algorithm 3 assumes $\eta$ is given. When $\eta$ is not know, we can run Algorithm 3 iteratively with a guess for $\eta$ and double the guess each iteration. We initialize by guessing that $\eta = 1$. Run Algorithm 3 on an auxiliary graph, which is just the residual graph plus a new source node $s^*$ with a single edge $(s^*, s)$ of capacity $\eta$ that is saturated upon initialization. If Algorithm 3 returns the cut $(s^*, s)$, we have augmented the pseudo-flow $f$ with an $s - t$ flow of $\eta$, but have not found an $s - t$ cut yet. Double $\eta$, change both the capacity and flow on $(s^*, s)$ to be the new $\eta$, and run Algorithm 3 again. Repeat this until we have found an $s - t$ cut. The initial $f$ is between $[\frac{\eta}{2}, \eta]$ away from being cut-saturating, and the whole procedure has $O(\eta \cdot n^2)$ running time. See Algorithm 5 for more details.

Summarizing this section, we prove our main theorem.

*Proof of Theorem 1.* Given a predicted pseudo-flow $\hat{f}$ with error $\eta$ on network $G$, Lemma 8 proved that Algorithm 5 finds a cut-saturating pseudo-flow $f$ for $G$ with error $\eta$ in time $O(\eta \cdot n^2)$. To find a min-cut, Lemma 5 shows that it is enough to find a pseudo-flow saturating a cut so that the $t$-side of the cut contains no excess and the $s$-side of the cut contains no deficit.

By Lemma 7, we can run Algorithm 3 seeded with $f$ on $G$ to obtain an updated cut-saturating pseudo-flow with no excess on the $t$-side of the maintained cut in time $O(\eta \cdot n^2)$. Then, Algorithm 3 can be run on the backwards network $B$, and from Corollary 2, the updated cut-saturating pseudo-flow now has no excess on the $t$-side of the cut and no deficit on the $s$-side.

The last phase can be left out if only the min-cut is desired. Suppose the min-cut is $\delta(S, T)$. By the proof of Lemma 5, to obtain a max-flow we only need to send all excess flow back to $s$, and send flow from $t$ to every deficit node. Label all nodes in $S$ with height $n$ and all nodes in $T$ with height 0. Then run Algorithm 2 to fix all excess in $S$. The algorithm will only send flow back to $s$, since there is no way to cross the cut $\delta(S, T)$. Then reverse the graph and flow, and run Algorithm 2 to fix the excess in the reversed graph, which exactly correspond to the deficit nodes in the original graph. □

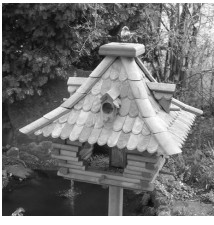 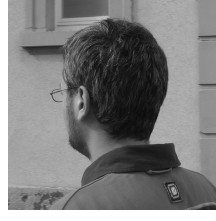 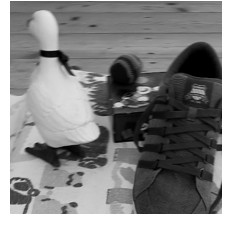 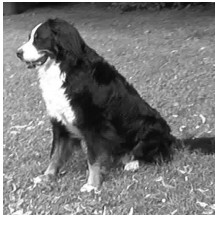

(a) Birdhouse        (b) Head        (c) Shoe        (d) Dog

Figure 2: The cropped and gray-scaled images from Figure 4 (copy from Figure 2 in [DMVW23]).

## 4 Empirical Results

In this section, we validate the theoretical results in Sections 3. To demonstrate the effectiveness of our methods, we consider *image segmentation*, a core problem in computer vision that aims at separating an object from the background in a given image. It is common practice to re-formulate image segmentation as a max-flow/min-cut optimization problem (see for example [BJ01, BK04, BFL06]), and solve it with combinatorial graph-cut algorithms.

The experiment design we adopt largely resembles that in [DMVW23], which studied warm-starting the Ford-Fulkerson algorithm for max-flow/min-cut. As in previous work, we do not seek state-of-the-art running time results for image segmentation. Our goal is to show that on real-world networks, warm-starting can lead to significant run-time improvements for the Push-Relabel min-cut algorithm, which claims stronger theoretical worst-case guarantees and empirical performance than the Ford-Fulkerson procedures. We highlight the following:

- Our implementation of cold-start Push-Relabel is much faster than Ford-Fulkerson on these graph instances, enabling us to explore the effects of warm-starting on larger image instances. This improved efficiency results from implementing the gap labeling and global labeling heuristics, both known to boost Push-Relabel's performance in practice. The actual running time scales with $\eta$ better than the theoretical $O(n^2\eta)$ bound. This is not totally surprising, as Push-Relabel is known to often enjoy subquadratic running time despite the bound.

- As we increase the number of image pixels (i.e., the image's resolution), the size of the constructed graph increases and the savings in time becomes more significant.

- Implementation choices (such as how to learn the seed-flow from historical graph instances and their solutions) that make the predicted pseudo-flow cut-saturating and that reroute excesses and deficits are crucial to the efficiency of warm-starting Push-Relabel.

**Datasets and data prepossessing** Our image groups are from the *Pattern Recognition and Image Processing* dataset from the University of Freiburg, and are titled BIRDHOUSE, HEAD, SHOE, and DOG. The first three groups are .jpg images from the *Image Sequences*[1] dataset. The last group, DOG, was a video that we converted to a sequence of .jpg images from the *Stereo Ego-Motion*[2] dataset.

Each of the image groups consists of a sequence of photos of an object and its background. There are slight variations between consecutive images in a sequence, which are the result of the object and background's relative movements or a change in the camera's position. These changes alter the solution to the image segmentation problem, but the effects should be minor when the change between consecutive images is minor. In other words, we expect an optimal flow and cut found on an image in a sequence to be a good prediction for the next image in the sequence.

From each group, we consider 10 images and crop them to be either $600 \times 600$ or $500 \times 500$ pixel images, still containing the object, and gray-scale all images. We rescale the cropped, gray-scaled images to be $N \times N$ pixels to produce different sized datasets. Experiments are performed for $N \in \{30, 60, 120, 240, 480\}$. In the constructed graph, we have $|V| = N^2 + 2$. Every graph is sparse, with $|E| = O(|V|)$, hence both $|V|$ and $|E|$ grow as $O(N^2)$. Detailed description of raw data and example original images can be found in Appendix B (Table 3, Figure 4).

---

[1]https://lmb.informatik.uni-freiburg.de/resources/datasets/sequences.en.html
[2]https://lmb.informatik.uni-freiburg.de/resources/datasets/StereoEgomotion.en.html

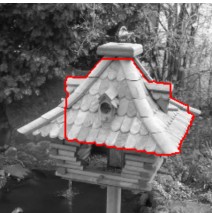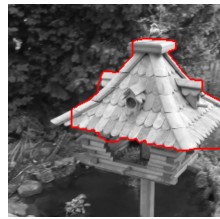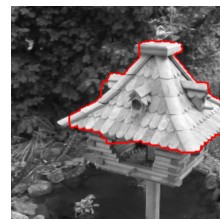

Figure 3: Cuts (red) on images chronologically evolving from the $240 \times 240$ pixel images from BIRDHOUSE.

**Graph construction**  As in [DMVW23], we formulate image segmentation as a max-flow/min-cut problem. The construction of the network flow problem applied in both our work and theirs is derived from a long-established line of work on graph-based image segmentation; see [BFL06]. The construction takes pixels in images to be nodes; and a penalty function value which evaluates the contrast between the pigment of any neighboring pixels to be edge capacity. We leave details on translating the images to graphs on which we solve max-flow/min-cut to Appendix B.

**Implementation details in warm-start Push-Relabel**  Throughout the experiments, whenever the Push-Relabel subroutine is called on any auxiliary graph, it is implemented with the gap relabeling heuristic, as shown in Algorithm 2, and the *global relabeling* heuristic, which occasionally updates the heights to be a node's distance from $t$ in the residual graph. These heuristics are known to improve the performance of Push-Relabel. As a tie-breaker for choosing the next active node to push from, we choose the one with highest height, which is known to improve the running time of Push-Relabel. We found the generic Push-Relabel algorithm without these heuristics to be slower than Ford-Fulkerson.

All images from the same sequence share the same seed sets. The constructed graphs are on the same sets of nodes and edges, but the capacities on the edges are different. The first image in the sequence is solved from scratch. For the second image in the sequence, we reuse the old optimal flow and cut from the first image one, then for the $i^{th}$ image in the sequence, we reuse the optimal flow and cut from the $i - 1^{st}$ image. We reuse the old max-flow on the new network by rounding down the flow on edges whose capacity has decreased, hence producing excesses and deficits, and pass this network and flow to the warm-start Push-Relabel algorithm in Section 3.

To find a saturating cut, instead of sending flow from $s$ to $t$ as suggested in Algorithm 5, we reuse the min-cut on the previous image $\delta(S_0, T_0)$ and send flow from $S_0$ to $T_0$ that originates from either $s$ or an excess node, and ends at either $t$ or a deficit node. We experimented with a few different ways of projecting the old flow to a cut-saturating one on the new graph. The way we implemented was by far the most effective, although it shares the same theoretical run-time as Algorithm 5.

The graph-based image segmentation method finds reasonable object/background boundaries. Figure 3 shows an example of how the target cut could evolve as the image sequence proceeds. Even with the same set of seeds, the subtle difference in images could lead to different min-cuts that need to be rectified. However, the hope is that the old min-cut bears much resemblance to the new one, hence warm-starting Push-Relabel with it could be beneficial. See Appendix B for other examples.

In our experiments, $\eta$ is estimated by computing both the total excess/deficit. Typically this is a loose bound, but it suffices for our purpose.

**Results**  Table 1 shows average running times for both Ford-Fulkerson in [DMVW23] and Push-Relabel. The "N/A" marks overly long run-time ($>1$ hour), at which point we stop evaluating the exact run-time.

These results show warm-starting Push-Relabel, while slightly losing in efficiency on small images, greatly improves in it on large ones. As for the scaling of run-time with growing data sizes, both cold- and warm- start's running time increases polynomially with the image width $n$, but warm-start scales better, and as $n$ increases to $480$, it gains a significant advantage over cold-start. Despite the different warm-start theoretical bounds ($O(\eta|V|^2)$ for Push-Relabel versus $O(\eta|E|)$ for Ford-Fulkerson), in practice both warm-start algorithms scale similarly as the dataset size grows. Additionally, one can see Push-Relabel greatly outperforms on the same image size, allowing us to collect run-time

Table 1: Average run-times (s) of cold-/warm-start Ford Fulkerson (FF) and Push-Relabel (PR)

| Image Group | FF cold-start | FF warm-start | PR cold-start | PR warm-start |
|---|---|---|---|---|
| BIRDHOUSE $30 \times 30$ | 0.80 | 0.51 | 0.05 | 0.06 |
| HEAD $30 \times 30$ | 0.62 | 0.43 | 0.05 | 0.05 |
| SHOE $30 \times 30$ | 0.65 | 0.39 | 0.07 | 0.06 |
| DOG $30 \times 30$ | 0.69 | 0.32 | 0.10 | 0.11 |
| BIRDHOUSE $60 \times 60$ | 8.22 | 3.25 | 0.30 | 0.45 |
| HEAD $60 \times 60$ | 9.36 | 4.10 | 0.50 | 0.50 |
| SHOE $60 \times 60$ | 8.09 | 3.04 | 0.69 | 0.47 |
| DOG $60 \times 60$ | 21.91 | 6.73 | 0.76 | 0.95 |
| BIRDHOUSE $120 \times 120$ | 109.06 | 37.31 | 5.42 | 4.98 |
| HEAD $120 \times 120$ | 101.79 | 28.43 | 5.90 | 5.92 |
| SHOE $120 \times 120$ | 98.95 | 30.44 | 6.44 | 3.74 |
| DOG $120 \times 120$ | 190.36 | 38.08 | 6.76 | 6.38 |
| BIRDHOUSE $240 \times 240$ | NA | 400.19 | 60.67 | 55.68 |
| HEAD $240 \times 240$ | NA | 374.79 | 32.46 | 31.00 |
| SHOE $240 \times 240$ | NA | 338.05 | 69.29 | 35.57 |
| DOG $240 \times 240$ | NA | 459.48 | 73.76 | 52.42 |
| BIRDHOUSE $480 \times 480$ | NA | NA | 604.54 | 502.58 |
| HEAD $480 \times 480$ | NA | NA | 365.25 | 285.75 |
| SHOE $480 \times 480$ | NA | NA | 756.77 | 364.42 |
| DOG $480 \times 480$ | NA | NA | 834.63 | 363.41 |

statistics on images of sizes up to $480 \times 480$ pixels, which we could not do with implementations of Ford-Fulkerson, due to its slow run-time.

Table 2 shows how the running time of warm-start Push-Relabel breaks down into the three phases described in Section 3: (1) finding a cut-saturating pseudo-flow; (2) fixing excess on $t$-side; (3) fixing deficits on $s$-side. Note phase (1) takes the most time, but results in a high-quality pseudo-flow, in that it takes little time to fix the excess/deficits appearing on the "wrong" side of the cut.

Table 2: Running time of warm-start Push-Relabel break down, on BIRDHOUSE

| Size | $30 \times 30$ | $60 \times 60$ | $120 \times 120$ | $240 \times 240$ | $480 \times 480$ |
|---|---|---|---|---|---|
| Total | 0.06 | 0.45 | 4.98 | 55.68 | 502.58 |
| Saturating cut | 0.04 | 0.34 | 4.17 | 46.25 | 431.49 |
| Fixing $t$ excesses | 0.01 | 0.09 | 0.53 | 5.29 | 64.01 |
| Fixing $s$ deficits | 0.01 | 0.02 | 0.27 | 4.13 | 7.08 |

## 5   Conclusions

We provide the first theoretical guarantees on warm-starting Push-Relabel with a predicted flow, improving the run-time from $O(m \cdot n^2)$ to $O(\eta \cdot n^2)$. Our algorithm uses a well-known heuristics in practice, the gap relabeling heuristic, to keep track of cuts in a way that allows for provable run-time improvements. One direction of future work is extending the approaches in this work to generalizations of $s$-$t$ flow problems, for instance, tackling minimum cost flow or multi-commodity flow. A different line of work is to develop rigorous guarantees for other empirically proven heuristics by analyzing them through a lens of predictions, providing new theoretical insights and developing new algorithms for fundamental problems.

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

# A More Discussion on Warm-starting Push-Relabel

We include some more insights and details on warm-starting Push-Relabel, including omitted proofs and algorithms from the main body.

## A.1 Omitted algorithms

Algorithm 4 is used throughout the paper to obtain valid heights on a cut-saturating pseudo-flow. It is essentially just a BFS procedure from the sink, where nodes that can reach the sink have height equal to their shortest distance to the sink in the residual graph, and nodes that cannot reach the sink have height $n$.

---

**Algorithm 4** Define Heights

---

    **Input**: Network $G$, a cut-saturating pseudo-flow $f$
    Define $h(s) = n$, $h(t) = 0$
    Run BFS in $G_f$
    Let $T = \{u \in V : \exists u - t \text{ path in } G_f\}$
    Let $S = V \setminus T$
    **for** all $u \in S$ **do**
        Let $h(u) = n$
    **for** all $u \in T$ **do**
        Let $h(u)$ be the shortest path length from $u$ to $t$ in $G_f$
    **Output**: Valid heights $h$, and cut parts $S$ and $T$

---

The following is a straightforward pre-processing algorithm. Given a pseudo-flow $\hat{f}$, it finds a cut-saturating pseudo-flow and computes the value of the error $\eta$.

---

**Algorithm 5** Find a Cut-saturating Pseudo-flow and Compute $\eta^*$

---

    **Input**: Network $G$, a pseudo-flow $\hat{f}$
    Initialize $\eta = 1$
    Build $G'$: a copy of $G$ plus a super-source $s^*$ and edge $(s^*, s)$
    Initialize $C = \delta(s^*, V)$
    **while** $C = \delta(s^*, V)$ **do**
        Update the capacity of $(s^*, s)$ in $G'$ to be $\eta$, and let $\hat{f}_{(s^*,s)} = \eta$
        Run Algorithm 3 with inputs $G'$ and $f^{\text{init}} = \hat{f}$, call output flow $f'$ and output cut $\delta(S, T)$
        Update $C \leftarrow \delta(S, T)$
        **if** the current cut $C = \delta(s^*, V)$ **then**
            Update $\eta \leftarrow 2\eta$.
    Let $\eta = f_{(s^*,s)}$, delete $s^*$ and $(s^*, s)$ from $G$.
    **Output**: $f$ and $\eta$

---

The mirror version of Algorithm 3 is below.

## A.2 Omitted proofs

*Proof of Lemma 3.* Fix $u \in S$, $v \in T$. Since $v$ can reach $t$ and $u$ cannot, any edge $(u, v)$ from $S$ to $T$ in $G$ must be saturated by $f$, and any edge $(v, u)$ from $T$ to $S$ in $G$ must have no flow. This is because if either of these were not true, the edge $(u, v)$ in $G_f$ would have positive capacity, allowing $u$ to reach $t$. Hence $\delta(S, T)$ is saturated by $f$. $\qquad\square$

*Proof of Lemma 4.* It is known from the proof of the vanilla Push-Relabel algorithm that all excess nodes in a pre-flow must have a path back to $s$; see Lemma 1. When $f$ saturates $\delta(S, T)$, such a path cannot go from $S$ to $T$, so the path must be within $S$. The last two lines of Algorithm 2 will resolve the excesses without effecting the saturated cut. So we have a feasible flow saturating a cut, meaning the flow is a max-flow and the cut is a min-cut. $\qquad\square$

---

**Algorithm 6** Moving all deficit to the $t$-side of the cut

---

**Input**: Network $G$, a pseudo-flow $f$ with error $\eta$ that saturates the cut $\delta(S_0, T_0)$
Build the residual $G_f$
Build $G'$ on copy of $G_f[S_0]$ plus $\{s^*, t^*\}$
**for** excess node $u \in S_0$ **do**
    Add edge $(s^*, u)$ with capacity $\mathsf{exc}_f(u)$
**for** deficit node $v \in S_0 \setminus \{s\}$ **do**
    Add edge $(v, t^*)$ with capacity $\mathsf{def}_f(u)$
Add edge $(s^*, s)$ with capacity $\eta + 1$ (or other number sufficiently large)
Let $f^{\text{init}}_{(s^*, u)} = c_{(s^*, u)}$ for all $(s^*, u)$ and $f^{\text{init}}_{(s^*, s)} = c_{(s^*, s)}$, and all other $f^{\text{init}}_e = 0$
Run Algorithm 2 on $G'$ and $f^{\text{init}}$, outputs $f'$ and $S'_0, S''_0$
**for** all copies of $e = (u, v) \in E(G_f)$ where $f'_e > 0$ **do**
    Update $f_e \leftarrow f_e + f'_e$
**Output**: Flow $f$ and cut parts $S'_0$ and $S''_0 \cup T_0$

---

*Proof of Theorem 2.* The algorithm first works to resolve excess in $T$, possibly moving nodes from $T$ to $S$ to do so. Once all excess is in $S$, correctness follows from Lemma 4. Note that the conditions of Lemma 4 are satisfied since by Lemma 3 the cut output by Algorithm 4 is saturated by $f$.

To bound the running time of Algorithm 2, we use a potential function argument that is different from that in the standard Push-Relabel analysis.

We first bound the running time of the main WHILE loop that terminates when all excess is contained in $S$ and the min-cut is found. We define the potential function $\Phi(T) = \sum_{u \in T} \mathsf{exc}_f(u) \cdot h(u)$. The operations involved change the value of $\Phi(T)$ in the following way.

– Saturated/Unsaturated push: In either case, at least one unit of excess flow is pushed from a higher height to a lower height, since for edge $(u, v)$ to be admissible, $h(u) = h(v) + 1$. Therefore, $\Phi(T)$ decreases by at least 1.

– Relabeling a node to open up new admissible edges: Any such relabeling operation increases $\Phi(T)$. However, the total of all of these increases is at most $\eta \cdot n^2$. The $\eta$ term upper bounds the possible excess at any node, whereas the $n^2$ term is because any node's (of which there are at most $n$) height only ever increases, and the height cannot increase beyond $n$ before it must leave $T$ permanently.

– Removing a node from $T$ by relabeling it to $n$: Decreases $\Phi(T)$.

Note that upon each operation of relabeling, the additional cost related to detecting the threshold value $\theta$ is $O(1)$ since we can maintain, for each value between 0 and current $\theta$, the number of nodes labeled with this height. Relabeling can only break the consecutive series if a node is the *only* node with this height value; hence this value will become the new $\theta$ and we remove from $T$ all nodes with heights above it.

Hence the total running time before finding the min-cut is bounded by $O(\eta \cdot n^2)$.

To bound the time for finding the max-flow, notice that the total excess in $G$ only decreases, so when we start to route excesses in $S$ to $s$, the total excess is also bounded by $\eta$. The same potential function argument can be used to prove it also takes $O(\eta \cdot n^2)$ time to resolve all excess in $S$, though using the fact that in Push-Relabel, heights are always bounded by $2n$ (see Lemma 1). $\qquad\square$

*Proof of Corollary 1.* Create an auxiliary graph $G'$ by taking a copy of $G$ and adding a super-source $s^*$ and an edge $(s^*, s)$ with capacity $\eta$. Create a pre-flow $f^{\text{init}}$ on $G'$ by saturating $(s^*, s)$ and letting $f_e = 0$ on all other edges in $G'$. Now run Algorithm 2 with inputs $G'$ and $f^{\text{init}}$. The initial (and maximum) excess in $G'$ was $\eta$, and so the run-time is bounded by $O(\eta \cdot n^2)$, as in the proof of Theorem 2. $\qquad\square$

*Proof of Corollary 2 .* We will build the reverse network of $G$, call it $B$ (for backwards). The network $B$ consists of a copy of $G$ but all of the edges go the opposite direction. More specifically, for every node $u \in V(G)$ there is a mirror node $u'$ in $B$, and for every edge $e = (u, v) \in E(G)$ with capacity

$c_e$, there is a mirror edge $e' = (v', u') \in E(B)$ with capacity $c_e$. Note that the source $s$ in $G$ is mirrored to the sink $s'$ in $B$, whereas the sink $t$ in $G$ is mirrored to the source $t'$ in $B$.

We can reverse any pseudo-flow $f$ on $G$ to be another pseudo-flow $f'$ on $B$, where for all $e \in E(G)$, $f'_{e'} = f_e$. Notably, $f$ and $f'$ saturate the same cut, and we observe $\mathsf{exc}_f(u) = \mathsf{def}_{f'}(u')$ and $\mathsf{def}_f(u) = \mathsf{exc}_{f'}(u')$.

Suppose we have a pseudo-flow $f$ that saturates cut $\delta(S_0, T_0)$ in $G$ with no excess nodes in $T_0$. Then in the backwards network $B$, $f'$ saturates $\delta(T_0', S_0')$, where $T_0'$ (resp. $S_0'$) is all mirror nodes $p'$ for such $p \in T_0$ (resp. $S_0$). Now $S_0'$ becomes the sink-side of the cut. In $B$, we can send flow from excess nodes and $s'$ to deficit nodes within $S_0'$, and this can be done by running Algorithm 3 on $B$.

The true algorithm in $G$ is Algorithm 6, as it is just the mirror image of Algorithm 3, though we may skip the execution of Algorithm 4, as we already know the cut. This flow, when reversed back into $G$, is the maximum amount of flow that can go from excess nodes and $s$ to deficit nodes in $G_f[S_0]$. After adding this reversed flow to $f$, the result is a cut-saturating pseudo-flow for $G$, where there is no deficit on the $s$-side of the cut. Observe that there is no excess or deficit created on either side of the cut in the process. $\qquad\square$

## A.3 Pre-processing the pseudo-flows

Given a pseudo-flow, the first phase is to pre-process $\hat{f}$ into a cut-saturating pseudo-flow on $G$ and to compute $\eta$. See Algorithm 5.

We create the auxiliary graph $G'$ as in Algorithm 5, and then run the gap-relabeling Push-Relabel on $G'$ (together with the standard initializing pre-flow) to find a minimum cut between $s^*$ and $t$ and obtain a flow $f'$. Corollary 1 bounds the Push-Relabel run-time in this case. Adding $f'$ to $\hat{f}$ creates a cut-saturating pseudo-flow. The next lemma proves the output of this algorithm satisfies the desired properties and that the algorithm runs in time $O(\eta \cdot n^2)$.

**Lemma 8.** *Suppose $\hat{f}$ is a predicted pseudo-flow with unknown error $\eta$ for network $G$. Then Algorithm 5 computes $\eta$ and finds a cut-saturating pseudo-flow $f$ for $G$ with error $\eta$ in time $O(\eta \cdot n^2)$.*

*Proof.* In the residual graph $G_{\hat{f}}$, the min-cut is bounded by $\eta$, since it is at most $\eta$ far from being cut-saturating. Therefore, we can apply Corollary 1 to $G_{\hat{f}}$ and obtain an optimal flow $f'$ on $G_{\hat{f}}$ in $O(\eta \cdot n^2)$ running time. The flow we desire is $f_e = f'_e + \hat{f}_e$ for all $e \in E$. It is cut-saturating for $G$ by the optimality of $f'$ on $G_{\hat{f}}$. Further, it is a pseudo-flow since $f'$ does not have any excess or deficit in $G_{\hat{f}}$ and clearly $f'_e + \hat{f}_e \leq c_e$ for all $e \in E$.

We turn our attention to the error value computed by Algorithm 5. Let $\eta^*$ be the true unknown value.

Note that in the auxiliary graph $G'$ constructed in Algorithm 5, if the current value of $\eta$ can be sent from $s^*$ to $t$, there exists a $s - t$ flow of $\eta$ in the residual graph; meaning when Push-Relabel terminates the $s^* - t$ cut we will find is just the edge $\eta$. This is a certificate that $\eta^* \geq \eta$. We double values of $\eta$ until we find some value where the cut in $G'$ is no longer $(s^*, s)$. This is a certificate that $\eta^* \leq \eta$. Given an upper bound and lower bound on $\eta^*$, one can run binary search in this range and continue using the cut-based certificates to further limit the range in which $\eta^*$ lies, until it is found exactly.

$\qquad\square$

Notably, one can also run Algorithm 2 and terminate it upon finding the min-cut, in which case $f'$ will be a pre-flow on $G_{\hat{f}}$, and the resulting $f = f' + \hat{f}$ will have total excess bounded by $2\eta$. In fact, one can do this in other steps of the algorithm as well, if the goal is only to find a min-cut, and only lose an additional constant factor in the running time; see Appendix A.4. As discussed, in practice one may wish to use a predicted cut instead of finding a cut-saturating pseudo-flow as in Algorithm 5.

By Definition 1, if a pseudo-flow $\hat{f}$ is $\sigma$ far from cut-saturating it means augmenting it by another flow $f$ with value at most $\sigma$ can saturate some cut. Let this cut be $\delta(S, T)$. Another way to look at

this is, within $\hat{f}$, the total flow passing through the cut $\delta(S, T)$ satisfies:

$$\sum_{u \in S, v \in T} \hat{f}_{(u,v)} - \sum_{u \in S, v \in T} \hat{f}_{(v,u)} \geq \sum_{u \in S, v \in T} c_{(u,v)} - \sum_{u \in S, v \in T} c_{(v,u)} + \sigma.$$

Apart from solving max-flow in the residual graph to saturate this cut, there may be other options to create a cut-saturating pseudo-flow. For example, the $\eta$ bound on error does not directly tell us where this cut is. However, if a practitioner can "guess" a good enough cut $\delta(S, T)$ from past problem instances, such a pseudo-flow can also be obtained simply by saturating all edges $(u, v) \in \delta(S, T)$ and removing the flow on all backward edges. The downside is that such a practice will transfer the error on that particular cut to the total excess and deficit on nodes incident to the cut. Overall, there may be a trade-off where one can omit Algorithm 5 in lieu of using a predicted cut, but at the cost of having to fix more excess and deficit in later steps.

### A.4 Early termination of auxiliary Push-Relabel upon finding min-cut

We mentioned that one can choose to quit the Push-Relabel algorithm on auxiliary graphs whenever a cut is found. The resulting pseudo-flow, although violating flow conservation constraints, can still be added to the initial pseudo-flow. We give a brief analysis of how this effects the execution of the algorithm.

The pseudo-flow is constructed in three places:

1. In Algorithm 5, where we saturate a cut;
2. In Algorithm 3, where we push flow from $t$-side excess nodes to deficit nodes and $t$;
3. In Algorithm 6, where we push flow from $s$-side excess nodes and $s$ to deficit nodes.

Notice this simple fact:

**Claim 2.** *For pseudo-flows $f, f',$ and $f''$ where $f = f' + f''$ (without violating capacities constraints), we have:*

$$\sum_u (\textbf{exc}_f(u) + \textbf{def}_f(u)) \leq \sum_u (\textbf{exc}_{f'}(u) + \textbf{def}_{f'}(u)) + \sum_u (\textbf{exc}_{f''}(u) + \textbf{def}_{f''}(u))$$

In Step 1, Algorithm 2 starts with $f_{\text{init}}$ with excess $\eta$, hence the resulting pre-flow also has at most $\eta$ excess, and adding this pre-flow without restoring it to a max-flow may increase the excess by $\eta$. In Step 2, the initial flow in $G'$ also has total excess of at most $\sum_{u \in T_0} \textbf{exc}_{\hat{f}}(u) \leq \eta$, so at the end of Algorithm 3 the total excess also increases by this much. In Step 3, correspondingly the maximum increase is $\sum_{u \in S_0} \textbf{def}_{\hat{f}}(u) \leq \eta$. To sum up, early termination in the auxiliary networks after finding the min-cut increases the total error by $O(\eta)$, and therefore has the same run-time bound up to a constant factor.

## B More on Experiments

### B.1 Omitted Tables and Figures for Experiments

Table 3 contains a detailed description of each of the four image groups, their original size in the raw dataset, the cropped grey-scaled image size, the foreground/background they feature, etc.

Table 3: Image groups' descriptions (copy of Table 1 from [DMVW23])

| Image Group | Object, background | Original size | Cropped size |
|---|---|---|---|
| BIRDHOUSE | wood birdhouse, backyard | 1280, 720 | 600, 600 |
| HEAD | a person's head, buildings | 1280, 720 | 600, 600 |
| SHOE | a shoe, floor and other toys | 1280, 720 | 600, 600 |
| DOG | Bernese Mountain dog, lawn | 1920, 1080 | 500, 500 |

Figure 4 gives one example of raw images from each image group.

In the main body, Figure 3 shows examples of cuts found in some images from the BIRDHOUSE image sequence. Figure 5, 7, 6 show example cuts from the other image groups.

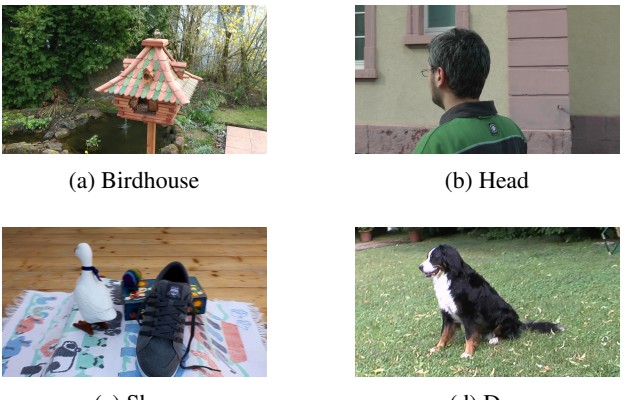

(a) Birdhouse                    (b) Head

(c) Shoe                         (d) Dog

Figure 4: Instances of images from each group (copy of Figure 1 from [DMVW23]).

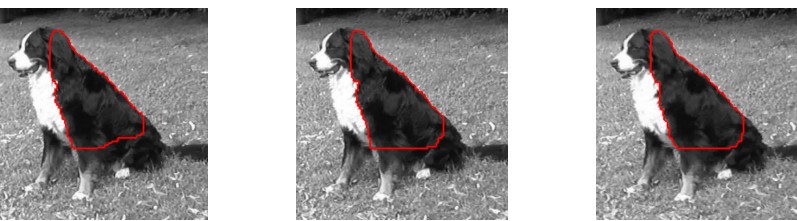

Figure 5: Cuts (red) on images chronologically evolving from the $240 \times 240$ pixel images from DOG.

## B.2   More On the Graph Construction

We take as input an image on pixel set $V$, and two sets of *seeds* $\mathcal{O}, \mathcal{B} \subseteq V$. The seed set $\mathcal{O}$ contains pixels that are known to be part of the object, while the seed set $\mathcal{B}$ contains pixels that are known to be part of the background. The *intensity* or gray scale of pixel $v$ is denoted by $I_v$. We say that two pixels are neighbors if they are either in the same column and in adjacent rows or same row and adjacent columns. Intuitively, if neighboring pixels have very different intensities, we might expect one to be part of the object and one to be part of the background. For any two pixels $p, q \in V$, a solution that separates them, i.e., puts one pixel in the object and the other one in the background, incurs a *penalty* of $\beta_{p,q}$. For neighbors $p$ and $q$, $\beta_{p,q} = C \exp(-(I_p - I_q)^2/(2\sigma^2))$, for $C$ a large constant, otherwise the penalty is 0. Note that the quantity $\beta_{p,q}$ gets bigger when neighbors $p$ and $q$ have stronger contrast.

A segmentation solution seeded with $\mathcal{O}$ and $\mathcal{B}$ labels each pixel as either being part of the object or part of the background, and the labeling must be consistent with the seed sets. Let $J$ denote the object pixels for a fixed segmentation solution. Then the *boundary-based* objective function is the sum of all of the penalties $\max_J \sum_{p \in J, q \notin J} \beta_{p,q}$, for $J$ with $\mathcal{O} \subseteq J, \mathcal{B} \subseteq V \setminus J$. As in the definition, a positive penalty cost is only incurred on the object's boundary. The goal is to minimize the total

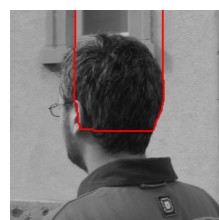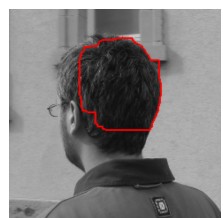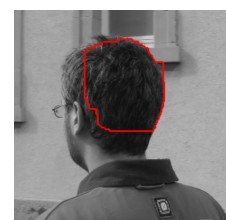

Figure 6: Cuts (red) on images chronologically evolving from the $240 \times 240$ pixel images from HEAD.

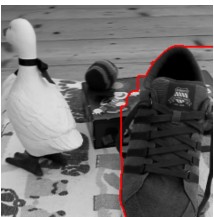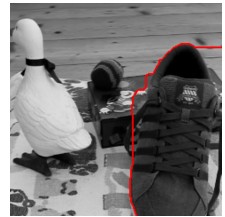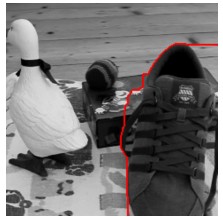

Figure 7: Cuts (red) on images chronologically evolving from the $240 \times 240$ pixel images from DOG.

penalty, which is in turn maximizing the contrast between the object and background, for the given object and background seed sets.

Solving this maximization problem is equivalent to solving the max-flow/min-cut problem on the following network. There is a node for each pixel, plus the object terminal $s$ and the background terminal $t$. As notation suggests, $s$ is the source of the network and $t$ is the sink. The edge set on the nodes is as follows: (1) for every $v \in \mathcal{O}$ add edge $(s, v)$ with capacity $M$, for $M$ a huge enough value that it is never saturated in any optimal cut; (2) for every $u \in \mathcal{B}$ add edge $(u, t)$, again with capacity $M$; (3) for every pair of nodes $p, q \in V$, add edges $(p, q)$ and $(q, p)$ with capacity $\beta_{p,q}$. If an image is on $n \times n$ pixels, note that the graph is sparse with $|V| = O(n^2)$ nodes and $|E| = O(n^2)$ edges.

In our experiments, all $\beta_{p,q}$'s are rounded down to the nearest integer, so that capacities are integral. Since $\beta_{p,q} \leq C$ by definition, it suffices for us to let $M = C|V|^2$.

## B.3    Other Settings

We have discussed how to cope with unknown $\eta$ in Algorithm 5 by using a doubling search from an initial guess $\eta = 1$ in order to guarantee the $O(\eta|V|^2)$ time bound. However, in practice, we have discovered that this is not absolutely necessary. One can use other surrogates for $\eta$ as long as they prove an effective upper bound on the error. In our experiments, $\eta$ is estimated by computing both the total excess/deficit induced by rounding down the flow, and the $\sigma$ (in the "$\sigma$ away from being cut saturating" notion) by using the old cut in the previous image in the sequence. Typically this is a loose bound, but it suffices for our purpose.

In these experiments the real error $\eta$ of the given predicted flow could be high. For example, on a sequence of 10 images of size $60 \times 60$ from the group DOG. The average total excess/deficit accounts for $69\%$ of the real min cut, whereas the $\sigma$, computed a posteriori using the min cut, accounts for about $23\%$. The results should be read together with the measured error metric value in [DMVW23] under the same experiment designs. Our $\eta$ is not negligible but still boosts Push-Relabel's performance with warm-starting, despite the dependency of the theoretical bound on $\eta$. This is typical — algorithms in practice often run much faster than their theoretical bounds. In [DMVW23], the empirical prediction errors (see, e.g., Table 5 in their Appendix) are at similar levels, and can be even higher than the max-flow value. As is discussed in their paper, the savings in running time results from the augmenting paths routing excess to deficit being much shorter than the source-sink augmenting paths; hence the performance goes beyond the theoretical $O(m\eta)$. Excesses/deficits are big in our setting, too, but the initialized pseudo-flow is already quite close to being cut-saturating. This might account for the speed-up we obtain.

