# OpenReview forum: "Warm-starting Push-Relabel"
_NeurIPS.cc/2024/Conference — NeurIPS 2024 poster_

### Official Review · Reviewer_D3Wp · 2024-06-26

**Soundness:** 2
**Presentation:** 2
**Contribution:** 3
**Rating:** 6
**Confidence:** 2

**Summary:**

This paper studies the push-relabel algorithm with warm-starts for the max-flow problem. Despite its empirical efficiency, the standard analysis of the algorithm is known to result in a somewhat pessimistic bound of $O(n^2m)$. The current paper considers a situation where the algorithm takes a predicted pseudo-flow $\hat f$ and has shown that the running time can be bounded by $O(n^2\eta)$, where $\eta$ is a prediction error that is at most $\\| \hat f - f^* \\|\_1$ for an optimal flow $f^*$. En route to this result, the authors have also shown that push-relabel with the gap-relabeling heuristic fed with a cut-saturating pre-flow (Algorithm 1) enjoys the same error-dependent bound. Overall, the algorithm converts $\hat f$ into a cut-saturating pseudo-flow $f$ in $O(n^2\eta)$ time (Algorithm 5), updates $f$ in $O(n^2\eta)$ time so that the $t$-side ($s$-side) has no excess (deficit) nodes (Algorithm 2), and fixes excess (deficit) in the $t$-side ($s$-side) with Algorithm 1. The main technical novelty lies in the update step, which swaps excess and deficit nodes between $t$- and $s$-sides. Finally, experiments on image-segmentation datasets confirm the effectiveness of push-relabel with warm-starts, compared with cold-start push-relabel and cold- and warm-start Ford--Fulkerson.

**Strengths:**

1. The push-relabel is an important practical algorithm with a large gap between theory and practice. The paper gives an interesting result to fill the gap.
2. Similarly, the analysis that applies to push-relabel with the gap-relabeling heuristic is also a nice result.
3. The algorithm for swapping excess and deficit nodes between $t$- and $s$-sides is an interesting new gadget.

**Weaknesses:**

1. I agree that analyzing push-relabel is important due to its practicality. On the theoretical side, however, the impact of the $O(n^2\eta)$ bound is somewhat weak since Ford-Fulkerson is already shown to run in $O(m \\| \hat f - f^* \\|\_1)$ time.
2. The paper has little implication about how we should learn $\hat f$. I understand that $\\| \hat f - f^* \\|\_1$ can be used as a surrogate loss that upper bounds $\eta$ in Definition 1. However, since it is mentioned in lines 86--87 that the $\ell_1$-error is not a good measure of the prediction quality, I would like to know whether we can develop a better method for learning $\hat f$ based on the error measure in Definition 1.
3. The paper is somewhat hard to follow as it involves many pseudo-codes, most of which are in the appendix. For me, the proof of Theorem 1, which includes a brief explanation of what each algorithm does, was helpful to overview the entire algorithm with predicted pseudo-flow. It might be better to present a similar technical overview when describing Figure 1.

**Questions:**

1. How can we know $\eta$ in Algorithm 2? Is it given as an output of Algorithm 5?
2. If my understanding is correct, in the experiments, $\hat f$ is set to the old max-flow in the previous image. Is this $\hat f$ attains small $\eta$?
3. I'm also curious about how the actual running time scales with the error level $\eta$ and how it aligns with the theoretical bound of $O(n^2\eta)$.


#### **Minor comments**
- Please define what "saturate" means for clarity.
- Do lines 64--65 and Lemma 3 state the same thing? If so, I would prefer to merge them or add a reference from lines 64--65 to Lemma 3.

**Limitations:**

Yes.

---

> ### Author Rebuttal · Authors · 2024-08-07
>
> Please see our responses to all reviewers for your questions on comparing warm start PR to warm start FF, as well as computing $\eta$.
>
> Comment: *I would like to know whether we can develop a better method for learning $\hat{f}$ based on the error measure in Definition 1.*
>
> Response: It is an interesting question as to whether our more fine-grained error metric can give insight into designing better predictions. We think there is room for improvement here. In our experiments, our predicted flow is the solution to a previous, similar instance. While these predictions can still have large error with respect to $\eta$, we believe it is on the right track to a more comprehensive error metric, which would measure how far $\hat{f}$ is to saturating a min cut. The literature on learning-augmented algorithms has not yet focused on obtaining predictions that are guaranteed to be of high quality with respect to the chosen error metric. A few works have started to explore this (see the recent work by Srinivas and Blum), but it is still nascent and seems quite challenging.
>
> Q: *If my understanding is correct, in the experiments, $\hat{f}$  is set to the old max-flow in the previous image. Does this $\hat{f}$ attain small $\eta$?*
>
> A: We refer to the comments addressed to all reviewers for a more detailed discussion. We have followed the practice in [Davies et al. 2023] when testing warm-start Ford-Fulkerson. Our $\eta$ contains both excess/deficit and the minimum slack from saturating any cut. In that sense, this choice of $\hat{f}$ does not attain small $\eta$, for rounding the previous max-flow by the new capacities results in a lot of excess/deficits. However, $\hat{f}$ is close to saturating the TRUE min-cut, which promises good initial progress for warm-starting.
>
> Q: *I'm also curious about how the actual running time scales with the error level $\eta$  and how it aligns with the theoretical bound of $O(n^2 \eta)$.*
>
> A: The actual running time scales with $\eta$ better than the theoretical $O(|V|^2 \eta)$. This is not totally surprising, as Push-Relabel is known to often enjoy subquadratic running time despite the $O(|V|^2|E|)$ bound. We refer to Table 2 for how the running time of warm-start Push-Relabel scales with growing image sizes. When the image size changes from $n \times n$ pixels to $2n \times 2n$, $|V|$ and $|E|$ grow by 4 since they are both roughly $n^2$, $\eta$ typically grows as fast as the max-flow/min-cut value, which in this case is a factor of 2. One can see that warm-start Push-Relabel’s actual performance goes beyond the theoretical bound, as does the cold-start Push-Relabel.

---

> > ### Comment · Reviewer_D3Wp · 2024-08-12
> >
> > Thank you for the detailed response. I appreciate your answers to my questions. I have increased my score to 6.

---

### Official Review · Reviewer_oTzZ · 2024-07-12

**Soundness:** 2
**Presentation:** 3
**Contribution:** 3
**Rating:** 7
**Confidence:** 3

**Summary:**

Authors provide the first theoretical guarantees for the Push-Relabel (PR) algorithm coupled with the gap relabeling heuristic, to address max flow problems, while using an arbitrary (pseudo-flow) initialization such as a predicted flow. The main result relates to proving a worse case complexity of $O(\eta * n^2)$ for their algorithm given a predicted pseudo-flow with error $\eta$ (cf Def 1) on the network with $n$ nodes. Finally, they validate empirically their theoretical analysis while showing the relevance of their algorithm compared to well-known competitive algorithms.

**Strengths:**

- Overall the paper is well-written.
- (Theorem 2) Authors provide the worse case complexity of their algorithm while using a cut-saturating pre-flow (arguably most intuitive init knowing maintains this structure along iterations)
- (Theorem 1) General case given any predicted pseudo flow.
- Good empirical validation of the theoretical results and comparison with FF algorithms using different warmstart strategy.

**Weaknesses:**

*edits after authors' rebuttal are in italic*

1. Potentially under-exploited empirical results:
   - a) *[addressed, see suggested improvements in discussions]*  visualizations corresponding to values reported to Table 2 / 5 could have been interesting to really observe the quadratic behaviour.
   - b) *[partly addressed in answer to all reviewers, paper to adjust accordingly]* Lack of analysis w.r.t the "parameter" eta, that relates to the error between the previously predicted flow and the new one. Potentially synthetic datasets could have been designed to control eta and compare theoretical results with empirical ones. It seems also possible on real-world dataset to compute a posteriori eta (e.g knowing all optimal flows over a sequence) along iterations to perform this comparison between theory and practice more accurately.
   - c) Even if authors explicitly state that their goal is not to claim SOTA results, it is interesting to have a benchmark to SOTA approaches for reference.

2. Limitations of theoretical results could be further discussed. In theory, FF algorithm has a better worse case complexity than the PR algorithm studied by authors, but the later is faster in practice. This either question the tightness of the bound given by authors, or that the average (not worse case) complexity is simply better for the later than the FF one, which is not theoretically studied by authors.

**Questions:**

I encourage authors to discuss and address the weaknesses I have highlighted above. A few questions follow for the sake of clarity:
- *[done]* Could you explain how the cold-start is performed ?
- *[done, to integrate in the paper]* How many times were run experiments to eliminate the biases of the state of the machine? I believe only once, so reporting averaged runtimes over several runs would be better. For reference, authors should also report the type of machines which were used to perform the experiments.

**Limitations:**

The authors have made an effort to address the limitations of their work, I encourage them to answer the questions above to ensure that all potential limitations have been covered.

No potential negative societal impact.

---

> ### Author Rebuttal · Authors · 2024-08-07
>
> Please see our responses to all reviewers for your questions on comparing warm start PR to warm start FF.
>
> Comment: *visualizations corresponding to values reported to Table 2 / 5 could have been interesting to really observe the quadratic behavior.*
>
> Response: Please see the pdf attached in the response to all reviewers (this was the only place we could attach a pdf). In a final version, we could take more time to find better scales for each dataset.
>
> Comment: *Lack of analysis w.r.t the "parameter" $\eta$, that relates to the error between the previously predicted flow and the new one.*
>
> Response: See the comments addressed to all reviewers for discussion on the a posteriori eta value measured on the real cuts. To sum up, we have a lot of excess/deficit resulting from rounding the previous max flow; however the pseudo-flow is close to saturating the real cut, making it a potentially good starting point for warm-start. The results match those in the evaluation of warm-start Ford-Fulkerson in [Davies et al., 2023] where prediction error is also big, but shorter augmenting paths used to fix excess/deficit lead to the improvement in warm-start.
>
> Q: *Could you explain how the cold-start is performed?*
>
> A: Based on each image from the sequence we construct a graph instance, where edges and their capacities are determined as instructed in Line 293-298. Then we run the vanilla Push-Relabel algorithm with gap relabeling (see Algorithm 1) on these graph instances individually. The algorithm itself is the standard Push-Relabel procedure that takes the graph, saturates all the edges going out of the source, labels the nodes with valid heights and finds an active node (in this case, the one with highest label among all the t-side nodes) to push flow from in every iteration.
>
> Q: *How many times were run experiments to eliminate the biases of the state of the machine? I believe only once, so reporting averaged runtimes over several runs would be better. For reference, authors should also report the type of machines which were used to perform the experiments.*
>
> A: The running time shown in Tables 1, 2, and 3 are averages taken over a sequence of 10 images for all 4 image groups to eliminate the bias of particular problem instances. The standard deviation is typically 5-15%, depending on different image groups and sizes. To eliminate the biases of machine status, repeating experiments on any fixed image instance has shown the per-image run-time standard deviation to be typically less than 1% of the average run-time, which is way less than the variance across different instances. This is measured over 100 repeated runs. If accepted, we will add the standard deviation data into the paper.
> All experiments are run on a device with processor Intel(R) Core(TM) i9-12900H @ 2.50 GHz, and 36GB memory.

---

> > ### Comment · Reviewer_oTzZ · 2024-08-12
> > **Answer to authors**
> >
> > Thank you for your detailed answers.
> >
> > Indeed for the visualizations it would be better to draw complete ranges of values in log-scale, with variances illustrated for each method with a lower intensity. Main points addressed in the answer to all reviewers should be clearly integrated in the paper.
> >
> > Overall I find authors' rebuttal compelling, therefore i increase my score from 6 to 7.

---

### Official Review · Reviewer_gRPX · 2024-07-15

**Soundness:** 3
**Presentation:** 3
**Contribution:** 3
**Rating:** 6
**Confidence:** 2

**Summary:**

This paper provides the running-time complexity analysis for warm-start Push-Relabel algorithm for the fundamental max-flow/min-cut problem. In particular, they study learning-augmented version of the Push-Relabel algorithm, where the algorithm can start from a pseudo-flow with error bounded by $\eta$. The main result is that a minimum cut can be found in time $O(\eta\cdot n^2)$, which improves upon the $O(n^2 m)$ runtime of basic Ford Fulkerson.

**Strengths:**

- This provides the first theoretical guarantees for warm-starting Push-Relabel with a predicted flow and demonstrates the benefits of learning-augmented version in improving the running time, where the predicted flow is close to an optimal flow.

- The theoretical results are further demonstrated by empirical experiments using image segmentation tasks over image sequences datasets, where the predicted flow is given by the result of the previous image sequence.

- The paper is well-written and well-structured, with clear motivations. The theoretical results deliver important insights about the practical benefits of utilizing learning-based predictions for the fundamental problems on graphs.

**Weaknesses:**

- The algorithm (Algorithm 2) requires a bound on the error $\eta$ of the prediction, which can be hard to estimate in practice.
- Even though the Push-Relabel performs much faster than Ford-Fulkerson procedures in image segmentation task (Sec. 4), when just comparing the theoretical bounds, $O(\eta n^2)$ (warm-starting Push-Relabel) and $O(\eta m)$ (warm-starting Ford-Fulkerson), the run-time benefits of warm-starting Push-Relabel cannot be seen compared to that of warm-starting Ford-Fulkerson. Is this due to a loose bound on the warm-starting Push-Relabel?

**Questions:**

- Theoretically, the provided run-time order for the warm-starting Push-Relabel is not smaller than that of the warm-starting Ford-Fulkerson) especially for a sparse graph when $m\ll n^2$, even though one can run Push-Relabel much faster empirically. Can the authors comment on any reasons for this result? Can the bound for warm-starting Push-Relabel be made even tighter by considering sparsity of the graphs?

**Limitations:**

No ethical limitations

---

> ### Author Rebuttal · Authors · 2024-08-07
>
> Please see our responses to all reviewers for your questions on comparing warm start PR to warm start FF, as well as computing eta.
>
> Comment: *The algorithm (Algorithm 2) requires a bound on the error $\eta$ of the prediction, which can be hard to estimate in practice.*
>
> Response: We refer the reviewer to our “Response for all reviewers” for a thorough discussion on how $\eta$ is computed and used. To sum up, $\eta$ can be estimated using the doubling search procedure described in Appendix A.3 (line 476-494) at no additional run-time complexity. The same procedure also transforms the given pseudo-flow to a cut-saturating one, both the flow and the estimated value $\eta$ are output together, and that $\eta$ is used as the input to Algorithm 2. In that response, we have also discussed $\eta$ in the experiments, and compared it to the min-cut. Note that despite the theoretical dependency on $\eta$, we obtained the efficiency improvements with non-negligible $\eta$ values. Hence we are reaching conclusions similar to that in [Davies et al, 2023].

---

> > ### Comment · Reviewer_gRPX · 2024-08-12
> >
> > Thank the authors for the detailed responses. I will keep my score.

---

### Official Review · Reviewer_C4XM · 2024-07-17

**Soundness:** 3
**Presentation:** 4
**Contribution:** 3
**Rating:** 7
**Confidence:** 2

**Summary:**

This paper propose and analyzes a warm starting scheme for the classical
push-relabel algorithm for max-flow problems. The basic approach is to convert
the warm-starting flow into a cut-saturating flow, at which point a clever
push-relabel scheme is used to ensure that the source-side of the cut has only
excess flow while the sink side of the cut has only flow deficits.  This
immediately yields a min-cut and is easily transformed into a max flow
solution.  The authors compare their approach with warm-starting methods for
the Ford-Folkerson algorithm, which they show is less efficient in practice.

**Strengths:**

- The paper is well written and understandable by non-experts.

- The warm-started version of push-relabel has a very nice instance-dependent
    complexity compared standard push-relabel ($O(m n^2)$ to $O(\eta n^2)$),
    allowing for fine-grained complexity estimates.

- Warm-started push-relabel works well in practice and is much faster than
    other warm starting methods for large problem instances.

**Weaknesses:**

- The error $\eta$ of a pseudo-flow $\hat f$ is hard to estimate without
    running Algorithm 5, meaning the total complexity of computation is not
    known until partway through execution. Moreover, this issue is not addressed
    in the paper.

- It is difficult to compare warm-started push-relabel with the Edmonds-Karp
    selection rule for Ford-Fulkerson because their complexities they depend on
    different aspects of the problem.

### Detailed Comments

Firstly, I want to say that this paper is significantly outside of my research
area (convex optimization) and so I cannot comment on the novelty of the ideas.
However, I think the ideas in this paper are of significant interest,
particularly for researchers focused on deriving instance-dependent
complexities. As such, I think the paper should probably be accepted.


**Computing $\eta$**:
Reading Section A.3, it seems like computing the value of $\eta$ is a
unaddressed difficulty in the main paper. I don't think it is reasonable to
consider $\eta$ to be input to the algorithm, so the doubling procedure in A.3
is required to estimate $\eta$ when running Algorithm 5. Since the complexity
of estimating $\eta$ with Algorithm 5 is no greater than that of running
Algorithm 5 with $\eta$ as in input, I think it makes sense to write the full
algorithm without knowledge of $\eta$ in appendix. Similarly, some comment
should be made about this issue in the main paper.


**Edmonds-Karp Selection Rule**:
You comment that Ford-Fulkerson with the Edmonds-Karp selection rule has
complexity $O(m \|f^* - \hat f\|_1)$. In comparison, warm started push-relabel
has complexity $O(\eta n^2)$. It is difficult to compare these two complexities
since $m \leq n^2$ but $\eta \leq \|f^* - \hat f\|_1$. In practice,
warm-started push-relabel seems to have much better complexity, but is it
possible to draw a rigorous comparison between the two methods? At the very
least, I think this issue should be addressed in the main paper.

**Questions:**

- Can $\eta$ ever exceed $O(m)$? That is, are the circumstances where
    the warm-started version of push-relabel has worse asymptotics than
    cold-start push relabel?


- What about using randomized predicted pseudo-flows $\hat f$ to
    warm-start push-relabel? I ask because, in continuous optimization, random
    initializations often out-perform default initializations like $0$. It
    would be interesting to choose a distribution for $\hat f$ and then compute
    the expect value of $\eta$ or perhaps high probability bounds on $\eta$.

**Limitations:**

Limitations are appropriately addressed in the paper.  The paper topic might be
considered somewhat niche to the majority of the NeurIPS audience. This is not
a weakness of the paper, however.

---

> ### Author Rebuttal · Authors · 2024-08-07
>
> Please see our responses to all reviewers for your questions on comparing warm start PR to warm start FF, as well as computing eta.
>
> Q: *Can $\eta$ ever exceed O(m)? That is, are the circumstances where the warm-started version of push-relabel has worse asymptotics than cold-start push relabel?*
>
> A: Indeed by definition \eta can be bigger than \Omega(m). However, the running time of our algorithm is never asymptotically worse than that of cold-start push relabel. This is because our algorithm solves Push-Relabel in several auxiliary graphs, and in each of them the running time is always bounded by the vanilla $O(n^2m)$ worst-case bound regardless of the min-cut value and $\eta$. In other words, the running time of our algorithm is actually $O(n^2 \min{m,\eta})$. We will update this in the final version.
>
> Q: *What about using randomized predicted pseudo-flows $\hat{f}$ to warm-start push-relabel? I ask because, in continuous optimization, random initializations often out-perform default initializations like $\vec{0}$. It would be interesting to choose a distribution for \hat{f}  and then compute the expected value of \eta or perhaps high probability bounds on $\eta$.*
>
> A: While a random $\hat{f}$ would likely have very large $\eta$, it would be interesting to study the expected value of $\eta$ given a distributional assumption on $\hat{f}$. The literature on learning-augmented algorithms has been content with proving the PAC-learnability of a prediction, and has not yet focused on obtaining predictions that are guaranteed to be of high quality with respect to the chosen error metric. A few works have started to explore this (see the recent work by Srinivas and Blum), but it is still nascent and seems quite challenging.

---

> > ### Comment · Reviewer_C4XM · 2024-08-08
> >
> > Thanks for your response and for the new plots.
> >
> > I think my concerns have been addressed in so far as that is possible. The difficulty of estimating $\eta$ before executing the algorithm is somewhat abated by the practical procedure described in the general response. The comparison to warm-started Ford-Fulkerson appears to be fundamental problem, at least until further technical tools are developed for analysis beyond the worst case.
> >
> > I will keep my score at 7 for now, but I do like this paper and feel it should be accepted.

---

### Author Rebuttal · Authors · 2024-08-07

We thank the reviewers for their careful reading of our paper! We are glad they agree that warm-starting the push relabel algorithm to find a max flow/min cut is an interesting theoretical question with potentially useful practical applications.

Please see comments on questions brought up by several reviewers here, and individual replies below each review. Also, the visualization requested by Reviewer oTzZ is attached.


**Comparing to work on warm-starting Ford Fulkerson**: Recall the cold-start running time of Edmonds Karp is $O(m^2n)$, while cold-start PR is $O(mn^2)$. Previous work in [Davies, et al. 2023] shows that warm-start FF has running time $O(m ||\hat{f}-f^*||_1)$, while we show that warm-start PR has running time $O(n^2 \eta) < O(n^2 ||\hat{f}-f^*||_1)$. Our theoretical result may not always be better than warm-start FF, depending on how close $\eta$ and $||\hat{f}-f^*||_1$ are (note Push-Relabel is less affected by sparsity then Edmonds Karp). Overall, this is not a weakness to our analysis, but is an artifact of the community’s  lack of understanding on why push-relabel performs so much better in practice than its theoretical bounds guarantee.

We further explain the theoretical importance of our result. Improving our theoretical understanding of algorithms that do well in practice (like Push-Relabel) is one of the pillars of the field of beyond worst-case analysis (most famously, the smoothed analysis to analyze the Simplex Method, though see also Tim Roughgarden's book and lectures on beyond worst-case analysis). Researchers have long suspected that there should be a way to parameterize instances or use practical heuristics to justify why Push Relabel is so much better in practice, and tools are actively being developed with a hope of eventually achieving this goal (e.g., the line of work by J. Diakonikolas and Song). Our work identifies that there is a monotonic property in the cuts when Push-Relabel is implemented with the gap relabeling heuristic that directly results in faster runtime when the value of the max flow/min cut is small (see our Corollary 1). While such a result was known for Ford-Fulkerson, this was NOT known for Push-Relabel prior to our work. We speculate this monotonic property may have further theoretical importance in improving Push-Relabel analyses on more general classes of networks.

We are happy to add this discussion to the main body of the paper in the technical contributions section.

**Computing $\eta$**: The error metric $\eta$, when unknown a priori, is computed synchronously while transforming the given pseudo-flow to be cut-saturating, and given as input to Algorithm 2 afterwards. This procedure is described in A.3, lines 476-491.  The doubling search finds $\eta$ “for free” (the found $\eta$ is at most twice the true error in Definition 1) without additional running time complexity for the overall algorithm. Note that the doubling trick is a standard tool in algorithm design for coping with unknown values.

We agree with a reviewer that more discussion on computing $\eta$ should be in the main body, hence we propose making the following changes to clarify how $\eta$ is found and used. In Appendix A.3, we will explicitly add pseudocode to formally describe the doubling search procedure for $\eta$: initialize with $\eta$=1; in each iteration call Algorithm 5 as a subroutine that takes $\eta$ as input and tries to augment the pseudo-flow by value $\eta$; if successful, then $\eta$ doubles. The procedure terminates upon saturating the flow, i.e., Algorithm 5 finds a $s-t$ cut instead of cutting the edge $(s^*, s)$. The new pseudocode, along with Algorithm 5, completes the pre-process algorithm for warm-start flow initialization, and also outputs a correct error bound $\eta$ that is used in subsequent push relabel operations. In the main body before Algorithm 2 is stated, we will clarify that $\eta$, if unknown, has to be found via the algorithmic approaches in A.3 and add a short summary on the doubling search procedure.

**Empirical Values of $\eta$**: Many reviewers are interested in the computation of $\eta$ in practice and its value in our image segmentation experiments. The doubling procedure in A.3 is an option, but one can use other surrogates for $\eta$ as long as they prove an effective upper bound on the error. In our experiments, $\eta$ is estimated by computing both the total excess/deficit, and the $\sigma$ (in the “$\sigma$ away from being cut saturating” notion in Line 75) by using the old cut in the previous image in the sequence. Typically this is a loose bound, but it suffices for our purpose.

We show the error metrics on a sequence of 10 images of size 60X60 from the group DOG. The average total excess/deficit accounts for 69% of the real min cut, whereas the $\sigma$, computed a posteriori using the min cut, accounts for about 23%. The results should be read together with the measured error metric value in [Davies, et al., 2023] under the same experiment designs. Our $\eta$ is not negligible but still boosts Push-Relabel's performance with warm-starting, despite the dependency of the theoretical bound on $\eta$. This is typical --- algorithms in practice often run much faster than their theoretical bounds. In [Davies, et al. 2023], the empirical prediction errors (see, e.g., Table 5 in the Appendix) are at similar levels, and can be even higher than the max-flow value. As is discussed in their paper, the savings in running time results from the augmenting paths routing excess to deficit being much shorter than the source-sink augmenting paths; hence the performance goes beyond the theoretical O(m \eta). Excesses/deficits are big in our setting, too, but the initialized pseudo-flow is already quite close to being cut-saturating. Perhaps this explains the speed-up we obtain.

---

### Decision · Program_Chairs · 2024-09-25

**Decision:**

Accept (poster)

**Comment:**

Dear authors -- thank you for an interesting well-written paper that provides new theoretical insights into the surprisingly good performance of warm-starting for push-relabel algorithms for max-flow.  While the topic of the paper is specialized to a fairly narrow area -- the reviewers all enjoyed the paper, and the analysis makes a tangible contribution to the "observed" (beyond worst case) behavior of combinatorial algorithms .  We hope that you'll be able to incorporate reviewer's suggestions in the final version (e.g. incorporating more detailed description of computing eta).  I recommend the paper for acceptance.